# Preferentially Paternal Origin of De Novo 11p13 Chromosome Deletions Revealed in Patients with Congenital Aniridia and WAGR Syndrome

**DOI:** 10.3390/genes11070812

**Published:** 2020-07-17

**Authors:** Tatyana A. Vasilyeva, Andrey V. Marakhonov, Natella V. Sukhanova, Sergey I. Kutsev, Rena A. Zinchenko

**Affiliations:** 1Research Centre for Medical Genetics, 115522 Moscow, Russia; vasilyeva_debrie@mail.ru (T.A.V.); kutsev@mail.ru (S.I.K.); renazinchenko@mail.ru (R.A.Z.); 2Central Clinical Hospital of the Russian Academy of Sciences, 119333 Moscow, Russia; natelasukhanova@gmail.com

**Keywords:** de novo chromosomal aberrations, gametogenesis, preferential parental origin, recombination, biased methylation, chromosomal breaks

## Abstract

The frequency of pathogenic large chromosome rearrangements detected in patients with different Mendelian diseases is truly diverse and can be remarkably high. Chromosome breaks could arise through different known mechanisms. Congenital *PAX6*-associated aniridia is a hereditary eye disorder caused by mutations or chromosome rearrangements involving the *PAX6* gene. In our recent study, we identified 11p13 chromosome deletions in 30 out of 91 probands with congenital aniridia or WAGR syndrome (characterized by Wilms’ tumor, Aniridia, and Genitourinary abnormalities as well as mental Retardation). The loss of heterozygosity analysis (LOH) was performed in 10 families with de novo chromosome deletion in proband. In 7 out of 8 informative families, the analysis revealed that deletions occurred at the paternal allele. If paternal origin is not random, chromosome breaks could arise either (i) during spermiogenesis, which is possible due to specific male chromatin epigenetic program and its vulnerability to the breakage-causing factors, or (ii) in early zygotes at a time when chromosomes transmitted from different parents still carry epigenetic marks of the origin, which is also possible due to diverse and asymmetric epigenetic reprogramming occurring in male and female pronuclei. Some new data is needed to make a well-considered conclusion on the reasons for preferential paternal origin of 11p13 deletions.

## 1. Introduction

Aniridia (OMIM #106210) (now called *PAX6*-associated aniridia syndrome [1,2]) is a dominantly inherited congenital panocular disorder which is caused by either heterozygous intragenic mutations in the *PAX6* gene (OMIM *607108) or heterozygous large chromosomal rearrangements of the 11p13 locus encompassing the *PAX6* gene or its distant regulatory elements. Large chromosomal deletions in the 11p13 region may include the *WT1* gene. About 40–60% of patients with deletions encompassing both the *PAX6* and *WT1* genes develop WAGR syndrome (characterized by Wilms’ tumor, Aniridia, and Genitourinary abnormalities as well as mental Retardation) (OMIM #194072) [3,4]. The frequency of gross chromosomal aberrations detected in patients with congenital aniridia (AN) in a world population is relatively high and can account up to 30% [5,6]. Theoretically, occurrence of chromosomal breaks in the 11p13 region could be explained by several known mechanisms of DNA strand break formation in normal cell during gametogenesis or later at the stage of the early zygote. They include breakage of non-B DNA structures, of DNA-RNA triplex formed during transcription [7,8], and of DNA loops within Topologically and Lamina Associated Domains (TADs and LADs) [9,10] as well as replication fork stalk-associated DNA damage [11] and recombination-based double strand DNA breaks [12].

To refine the breakage mechanism in the 11p13 region, one should take into account the nucleotide sequence at the break points and near them, the chromatin epigenetic state and its dynamics, a local chromatin architecture, and its dynamics because breaks could arise due to any of the reasons causing a mechanical stress and/or disorder of complex biological processes and interactions involving chromatin as well as to a failure of the breakage repair. Hence, even without the benefit to explore each of the issues mentioned above, one could attempt to make a well-considered suggestion on a putative 11p13 breakage mechanism based on the 11p13 deletions’ specific features established for a real cohort of patients. 

## 2. Materials and Methods 

Ten families with a proband with congenital aniridia (*n* = 9) or WAGR syndrome (*n* = 1) and de novo 11p13 chromosome deletions were included into the study. The mean age of the fathers was 28.8, with median 28.0 years (25–75% range: 27.0–29.8, min 25.0, max 37.0), and the mean age of the mothers was 28.5, with median 28.0 years (25–75% range: 27.3–30.3, min 23.0, max 34.0). The ages of the fathers and mothers do not differ (*p* = 0.72201, Student’s *t*-test). Parents’ ages and clinical picture of the probands are listed in a Appendix A.

Multiplex ligation-dependent probe amplification (MLPA) analysis was performed with SALSA MLPA probmix P219-B2 PAX6 (MRC-Holland, Amsterdam, The Netherlands) according to the manufacturer’s recommendations. The results of fragment analysis after MLPA reaction were analyzed using Coffalyser.Net (MRC-Holland). This probe mix covered chromosome 11 area chr11:27636398–35117389 (according to the NCBI36/hg18 assembly of human genome).

Linkage and loss of heterozygosity analyses (LOH) of short tandem repeat markers (STR) were implemented as described earlier [13]. STR markers set an enclosed area chr11:29898018–33144526 (according to the NCBI36/hg18).

Statistical analysis was performed usingbinomial probability calculator.

The clinical and molecular genetic study was performed in accordance with the Declaration of Helsinki and was approved by the Institutional Review Board of the Research Centre for Medical Genetics, Moscow, Russia, with written informed consent obtained from each participant and/or their legal representative as appropriate.

## 3. Results

The loss of heterozygosity (LOH) analysis based on short tandem repeat (STR) marker segregation pattern analysis was performed in 10 families with a proband with aniridia or WAGR syndrome and de novo 11p13 chromosome deletions. Localization of chromosome breakpoints of 0.9–7.5-Mb-long deleted regions varied widely within the studied genome region chr11:30632179-33144526 (NCBI36/hg18). STR analysis was not informative in 2 families. In 7 out of examined 8 families, the deletion occurred on the paternal allele. Thus, the following reasoning assumes that the paternal origin of de novo 11p13 chromosome deletions is not random (binomial probability of equiprobable distribution *p =* 0.03125) (Table 1). In case of familial transmission of pathogenic deletions of 11p13 loci, the distribution of origins was 6 maternal versus 3 paternal (*p =* 0.16406) (Figure 1).

## 4. Discussion

Frequent paternal origin of 11p13 de novo microdeletions is in agreement with existing views on the difference in de novo and familial chromosome rearrangement rates occurring on paternal and maternal alleles. Usually, familial balanced rearrangements are transmitted through a mother and de novo ones appear to be mainly of paternal origin [14]. For example, about 80% of de novo genome wide copy number variants (CNVs) associated with intellectual disability were shown to arise on the paternal haplotype [15]. CNVs could arise from several different mechanisms which are strongly defined by the chromatin architecture and which perhaps are peculiar for each disorder-associated genome locus [16]. Two types of most common deletions encompassing the *NF1* gene present an example of well-studied and totally different related mechanisms as well as parental bias in the deletion origin [17]. Nevertheless, most disease-associated CNVs are proposed to be nonrecurrent and arise via replication-based mechanisms [18]. 

The cause of the breakage in the 11p13 chromosome region as well as the difference in its parental origin are still unclear, though predominantly the paternal origin of 11p13 de novo microdeletions were described earlier [19]. Considering that loss of maternal 11p13 allele was typical for somatic mutation in Wilms tumors, the authors proposed that they should find loss of the paternal 11p13 allele as a germline primary mutation in patients with defined chromosome 11p13 deletions. Eventually, they determined that de novo 11p13 deletions occurred on the paternally derived chromosome in seven out of eight children [19]. 

Firstly, large rearrangements may arise on the chromosomes of different parental origins during gametogenesis due to recombination. Recombination breakage should affect both paternal and maternal alleles roughly with the same frequency. On the one hand, a quantity of cell divisions and frequency of recombination in spermatogenesis could be, at least, partially responsible for the observed paternal deletions predominance. Interestingly, histone methyltransferase PRDM9 (MEISETZ—meiosis-induced factor containing a PR/SET domain (PRDF1-RIZ (PR) homology domain sybtype of SET domain (Su(var)3-9, enhancer-of-zeste and trithorax)) and zinc finger motif, OMIM *609760), which activates hot spots for recombination [20], is expressed in female gonads just in the fetus and in the testis postnatally during the lifetime [21]. On the other hand, according to the Marshfield Comprehensive human genetic maps, the female recombination rate in the 11p13 region is about 2.14 times greater than that for males, suggesting enhanced meiotic recombination for this genome region in females [22]. Thus, the preferential paternal origin of de novo 11p13 deletions in patients with aniridia and WAGR could not be explained by recombination differences. Moreover, the great majority of structure variants are considered no longer to be formed through recombination [23]. 

Nevertheless, there are some other strong arguments in favor of the idea that DNA breakage could have occurred more frequently during spermatogenesis. Male germ cell development is a very specific process in view of chromatin state remodeling [24]. Spermatids undergo a unique process of chromatin reorganization and package into highly condensed state through histone-to-protamine replacement [25]. The chromatin transition is thought to be associated with DNA breaks which could facilitate DNA supercoiling elimination [26]. 

Another explanation of nonrandom loss of the paternal allele in aniridia patients could be based on an actual evidence that de novo deletions in 11p13 often occur in the postzygotic stage [5]. About 16% of de novo 11p13 deletions are mosaic, represented in 30–70% of the cells [27]. That is why these deletions are assumed to arise not in gametogenesis but later in the early postzygotic period [27]. Heterozygous deletions may have occurred on the chromosomes of a certain parental origin after fertilization but before the time of the loss of the parental epigenetic marks. That could have happened in a narrow time interval during a very early period of zygote development before cleavage. In this short timespan, asynchronous processes of epigenetic reprogramming of chromosomes of different parental origin occur [28]. Initially, paternal loci actively and rapidly lose CpG methylation before replication, and after that, maternal loci undergo passive demethylation during replication [29]. Demethylation in paternal DNA occurs via a base excision repair mechanism and is supposed to be linked to DNA breaks [30,31]. Stops of transcription spermatozoids have no mechanism to repair DNA damage [32]. Paternal DNA damage is also well insured by a maternal repairing machinery in a zygote after fertilization [33,34]. Thus, paternal chromatin integrity depends on the capacity of the oocyte to repair it [32]. Maternal base excision repair machinery is supposed to fix paternal DNA damage at the expense of its demethylation [33]. Therefore, there could be some selection pressure against zygotes with aberrant methylation in favor of that with a small lack of genome material during this stage; as a result, paternal deletions could slip through DNA lesion zygotic checkpoints. 

The 11p13 region chromatin stiffness and flexibility could be under a considered influence of the chromatin epigenetic features [9]. Chromatin remodeling during spermatogenesis and, after that, in paternal pronucleus in a zygote may be associated with DNA breaks and could serve as a possible explanation for preferentially paternal origin of the alleles with 11p13 deletions [35]. 

Finally, the predominance of de novo deletions on the paternal alleles also may have indicated possible imprinting of the loci of the deleted genes [36,37]. Maternal alleles with such a deletion may influence cell viability, either in gametes or zygotes. For congenital aniridia, the ratio of affected-to-normal offspring of an affected parent of either sex was defined to be 38 to 62 [38]. That could mean a decreased viability of cells with 11p13 deletions, regardless of parental origin of the pathogenic allele. It should be noted that some differentially methylated germline loci in males and females have been defined to be transiently imprinted. Methylation differences could be protected until the blastocyst stage as they may influence early preimplantation development [39]. On the other hand, the transmission ratio of the familial deletions (6 maternal versus 3 paternal) obtained here and in an earlier study rather denied the last imprinting-related suggestion [19]. 

## 5. Conclusions

Deletions in both spermatozoid and zygote DNA on paternal alleles are suggested to be associated with dramatic chromatin state fluctuations. The base excision repair mechanism promotes both loss of DNA supercoiling during histone-to-protamine replacement and active demethylation in paternal pronucleus. 

However, the observation of the predominantly paternal origin of the allele with 11p13 de novo deletions in patients with aniridia remains so far only an observation, the explanation of which requires more in-depth studies and the accumulation of additional data.

## Figures and Tables

**Figure 1 genes-11-00812-f001:**
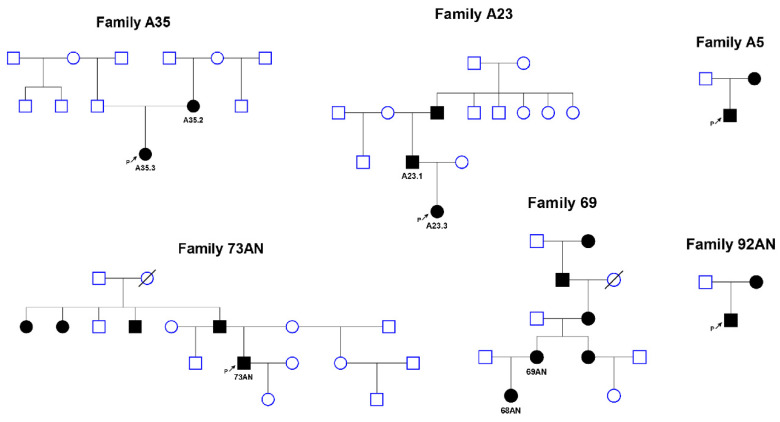
Pedigrees of familial cases of aniridia caused by deletions of the 11p13 region.

**Table 1 genes-11-00812-t001:** Patients with de novo deletions of 11p13 region.

Patient’s ID	Break Points Coordinates According to 11p13 MLPA	Genes Affected by Deletion	Origin ^1^
A-25	chr11:31824328–31832887	*PAX6ex1–PAX6ex5*	nd
A-36	chr11:31671656–32339851	*ELP4ex9–PAX6–RCN1*	pat
A-30	chr11:27679822–33374888	*BNDF−FSHB−DCDC1−ELP4–PAX6−RCN1−WT1−HIPK3*	pat
52.03	chr11:27679822–35160813	*BNDF−FSHB−DCDC1−ELP4–PAX6−RCN1−WT1−HIPK3−LMO2−EHF−CD44*	pat
20.03	chr11:30253552–32457265	*FSHB–DCDC1−ELP4−PAX6–RCN1−WT1*	pat
02.12	chr11:31329311–31671656	*DCDC1−ELP4ex9*	pat
09.03	chr11:31329311–31671656	*DCDC1−ELP4ex9*	nd
04.14	chr11:31391209–31838055	*DCDC1ex1−ELP4−PAX6int1*	mat
36.03	chr11:30253552–32125308	*PAX6ex7−RCN1*	pat
A-26	chr11:31329311–32339851	*DCDC1-ELP–PAX6–RCN1*	pat

^1^ Note: nd: the origin cannot be defined, pat: paternal, mat: maternal origin.

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
