# Peer review of "Preferentially Paternal Origin of De Novo 11p13 Chromosome Deletions Revealed in Patients with Congenital Aniridia and WAGR Syndrome"

_genes, 2020, doi:10.3390/genes11070812_

Round 1
Reviewer 1 Report
WAGR syndrome is a relatively rare condition that constitutes a subset of congenital aniridia cases. The paper by Vasilyeva et al. seeks to understand the origin of de novo chromosome deletions in WAGR syndrome. Deletions in 10 families were mapped and for 7 of the deletion was found to be have occurred on the paternal side. The authors are unable to draw any conclusions beyond this with their current findings. The information reported will be useful for a variety of researchers and should be made available.
Comments:
Is there any data on the age of the parents, especially the father?
No patient phenotypic information other than aniridia is reported, and this would be very useful for genotype-phenotype correlations. Is there any overlap between the patients studied and those reported in the author’s May 2020 paper in Journal of Medical Genetics? This reference should be included.
Presentation of results: Most papers of this nature have a chromosome diagram showing the extent of the different deletions. It would also be very helpful to have genetic landmarks relevant to WAGR syndrome such as those within (WT1, PAX6, DCDC1, PRRG4) and outside the critical region (BDNF, SLC1A2) on the diagram or summarized in the text/abstract. This would greatly increase the accessibility of the work to other researchers.
There is sufficient background in the introduction. The methods are straightforward and adequately described and statistical significance reported. Conclusions are supported by the data, although why de novo deletions occur at 11p13 is still unknown.
Author Response
Comments and Suggestions for Authors
WAGR syndrome is a relatively rare condition that constitutes a subset of congenital aniridia cases. The paper by Vasilyeva et al. seeks to understand the origin of de novo chromosome deletions in WAGR syndrome. Deletions in 10 families were mapped and for 7 of the deletion was found to be have occurred on the paternal side. The authors are unable to draw any conclusions beyond this with their current findings. The information reported will be useful for a variety of researchers and should be made available.
Comments:
Is there any data on the age of the parents, especially the father?
Response: The median age of the fathers was 28 years old, it did not differ from median mothers’ age (28 years old), we added the parental age into the Table in Supplementary table, and characteristics of the cohort – into the Materials and Methods section.
No patient phenotypic information other than aniridia is reported, and this would be very useful for genotype-phenotype correlations. Is there any overlap between the patients studied and those reported in the author’s May 2020 paper in Journal of Medical Genetics? This reference should be included.
Response: Patients phenotypic information we added to the data in the Table in Supplementary table, and all patients reported here were described earlier and were included into the study of genotype phenotype correlations. We have referred to our previous publications.
Presentation of results: Most papers of this nature have a chromosome diagram showing the extent of the different deletions. It would also be very helpful to have genetic landmarks relevant to WAGR syndrome such as those within (WT1, PAX6, DCDC1, PRRG4) and outside the critical region (BDNF, SLC1A2) on the diagram or summarized in the text/abstract. This would greatly increase the accessibility of the work to other researchers.
Response: We have updated Table 1 with affected genes, though the SLC1A2 gene is out of the tested region.
There is sufficient background in the introduction. The methods are straightforward and adequately described and statistical significance reported. Conclusions are supported by the data, although why de novo deletions occur at 11p13 is still unknown.
Reviewer 2 Report
The authors presents the manuscript “Preferentially paternal origin of de novo 11p13 chromosome deletions revealed in patients with congenital aniridia and WAGR syndrome”
Microdeletion and microduplication syndromes are multiple, this happens in other syndromes? there is a lack of literature review. The only cited manuscript is Pellestor, F.; Anahory, T.; Lefort, G.; Puechberty, J.; Liehr, T.; Hedon, B.; Sarda, P. Complex chromosomal rearrangements: Origin and meiotic behavior. Hum Reprod Update 2011, 17, 476-494.
Perhaps the search for these data will allow us to decide on one of the two theories presented by the authors: 1) Chromatin remodeling during spermatogenesis and after that in paternal pronucleus in a zygote may be associated with DNA breaks or 2) Imprinting of the loci of the deleted genes.
Publishing these data is interesting but it is necessary to refer to other studies to know if it is already described and that these data add a little more information.
Author Response
The authors presents the manuscript “Preferentially paternal origin of de novo 11p13 chromosome deletions revealed in patients with congenital aniridia and WAGR syndrome”
Microdeletion and microduplication syndromes are multiple, this happens in other syndromes? there is a lack of literature review. The only cited manuscript is Pellestor, F.; Anahory, T.; Lefort, G.; Puechberty, J.; Liehr, T.; Hedon, B.; Sarda, P. Complex chromosomal rearrangements: Origin and meiotic behavior. Hum Reprod Update 2011, 17, 476-494.
Perhaps the search for these data will allow us to decide on one of the two theories presented by the authors: 1) Chromatin remodeling during spermatogenesis and after that in paternal pronucleus in a zygote may be associated with DNA breaks or 2) Imprinting of the loci of the deleted genes.
Response: Thank you very much, that is true, but we are limited by the format of the short communication. Microdeletion and microduplication syndromes are really numerous, and CNVs may derive from several different mechanisms. Though, most disease-associated CNVs are nonrecurrent and arise via replication-based mechanisms PMID: 21846967. Nonrecurrent deletions in the 11p13 chromosome usually are not flanked by any repeats and are proposed to arise via breaks induced replication (and due to microhomology). Here we refer to the metanalysis of chromosome deletions breakpoints context based on 26927 CNVs from the 1000 Genomes Project PMID: 30773596. We added a paragraph on the requested data into the discussion section and the references in the list.
Publishing these data is interesting but it is necessary to refer to other studies to know if it is already described and that these data add a little more information.
Reviewer 3 Report
Review on manuscript: genes-853218-v1
In the presented manuscript authors Vasilyeva T. at al. analyzed the origin of de novo 11p13 chromosome deletions in patients with congenital aniridia and WAGR syndrome. The authors performed loss of heterozygosity analysis based on short tandem repeat (STR) marker segregation pattern analysis in 10 families with a proband with aniridia or WAGR syndrome and de novo 11p13 chromosome deletions. In two families, however, STR analysis was not informative and the analysis of the remaining 7 out of 8 families showed that the deletion occurred on the paternal allele. The authors therefore assume that the paternal origin of de novo 11p13 chromosome deletions is not random. Moreover, the familiar transmission of pathogenic deletions of 11p13 loci was 6 maternal vs. 3 paternal.
The manuscript is well written and the assumptions from the study are reasonable. In-depth studies, however, are needed to make a reasonable conclusion on preferentially paternal origin of 11p13 deletions.
There are few minor points with recommendations that authors would consider in order to improve the present manuscript before publishing:
1. Page 3, Line 83 …is in agreement with… instead of …is in agree with…
2. The Figure 1 is informative but please make sure it appears much larger in the final version. The present version is difficult to read.
3. It will be better if the first sentence in the first paragraph of Conclusion would not start with ‘Thus”
Author Response
In the presented manuscript authors Vasilyeva T. at al. analyzed the origin of de novo 11p13 chromosome deletions in patients with congenital aniridia and WAGR syndrome. The authors performed loss of heterozygosity analysis based on short tandem repeat (STR) marker segregation pattern analysis in 10 families with a proband with aniridia or WAGR syndrome and de novo 11p13 chromosome deletions. In two families, however, STR analysis was not informative and the analysis of the remaining 7 out of 8 families showed that the deletion occurred on the paternal allele. The authors therefore assume that the paternal origin of de novo 11p13 chromosome deletions is not random. Moreover, the familiar transmission of pathogenic deletions of 11p13 loci was 6 maternal vs. 3 paternal.
The manuscript is well written and the assumptions from the study are reasonable. In-depth studies, however, are needed to make a reasonable conclusion on preferentially paternal origin of 11p13 deletions.
There are few minor points with recommendations that authors would consider in order to improve the present manuscript before publishing:
- Page 3, Line 83 …is in agreement with… instead of …is in agree with…
Response: Corrected
- The Figure 1 is informative but please make sure it appears much larger in the final version. The present version is difficult to read.
Response: Thank you, corrected.
- It will be better if the first sentence in the first paragraph of Conclusion would not start with ‘Thus”
Response: Thank you, Conclusion has been corrected
Reviewer 4 Report
In this manuscript, Vasilyeva et al. aim to investigate the genetic mutations in patients with congenital aniridia and WAGR syndrome. The authors performed loss of heterozygosity analysis in 10 families with de novo deletions and found that 7 out of 8 informative families had mutations occurred on the paternal alleles. The authors proposed that the non-random loss of paternal allele may occur during gametogenesis due to recombination, or in the postzygotic stage.
The findings of the study are not new as the paternal origin of de novo mutations on 11p13 have been described a long time ago (e.g. Huff, V., et al. "Parental origin of de novo constitutional deletions of chromosomal band 11p13." American journal of human genetics 47.1 (1990): 155.). The conclusion of this manuscript is consistent with the previous studies.
Overall, the study is well-conducted, and the manuscript is nicely prepared. To make the manuscript more informative, I would suggest the authors include more clinical information about the patients, e.g. the age of onset and other clinical presentations.
Author Response
In this manuscript, Vasilyeva et al. aim to investigate the genetic mutations in patients with congenital aniridia and WAGR syndrome. The authors performed loss of heterozygosity analysis in 10 families with de novo deletions and found that 7 out of 8 informative families had mutations occurred on the paternal alleles. The authors proposed that the non-random loss of paternal allele may occur during gametogenesis due to recombination, or in the postzygotic stage.
The findings of the study are not new as the paternal origin of de novo mutations on 11p13 have been described a long time ago (e.g. Huff, V., et al. "Parental origin of de novo constitutional deletions of chromosomal band 11p13." American journal of human genetics 47.1 (1990): 155.). The conclusion of this manuscript is consistent with the previous studies.
Response: Thank you very much for the reference. We add the article in the list and add sentences in the discussion. A great thought was to search for germline deletions on the paternal alleles, if in tumor maternal alleles loss had been established! We also found there supporting information on the equal frequency of deleted alleles transmission from mothers and from fathers.
Overall, the study is well-conducted, and the manuscript is nicely prepared. To make the manuscript more informative, I would suggest the authors include more clinical information about the patients, e.g. the age of onset and other clinical presentations
Response: We added clinical information on the patients and the age of parents into the Supplementary table.